

# *MME* and *PTPRC*: key renal biomarkers in lupus nephritis

Min Wen, Marady Hun, Mingyi Zhao and Qingnan He

Department of Pediatrics, The Third Xiangya Hospital, Central South University, Changsha, Hunan, China

## ABSTRACT

**Background.** Lupus nephritis (LN) is an autoimmune-related kidney disease with a poor prognosis, however the potential pathogenic mechanism remains unclear and there is a lack of precise biomarkers. Therefore, a thorough screening and identification of renal markers in LN are immensely beneficial to the research on its pathogenic mechanisms and treatment strategies.

**Methods.** We utilized bioinformatics to analyze the differentially expressed genes (DEGs) at the transcriptome level of three clusters: total renal, glomeruli, and renal tubulointerstitium in the GEO database to discover potential renal biomarkers of LN. We utilized NephroSeq datasets and measured mRNA and protein levels in the kidneys of MRL/lpr mice to confirm the expression of key DEGs.

**Results.** Seven significantly differential genes (*EGR1, MME, PTPRC, RORC, MX1, ZBTB16, FKBP5*) were revealed from the transcriptome database of GSE200306, which were mostly enriched in the pathway of the hematopoietic cell lineage and T cell differentiation respectively by KEGG and GO analysis. The seven hot differential genes were verified to have consistent change trends using three datasets from NephroSeq database. The receiver operating characteristic (ROC) curve indicated that five DEGs (*PTPRC, MX1, EGR1, MME* and *RORC)* exhibited a higher diagnostic ROC value in both the glomerulus and tubulointerstitium group. Validation of core genes using MRL/lpr mice showed that *MME* and *PTPRC* exhibit significantly differential mRNA and protein expression patterns in mouse kidneys like the datasets.

**Conclusions.** This study identified seven key renal biomarkers through bioinformatics analysis using the GEO and NephroSeq databases. It was identified that *MME* and *PTPRC* may have a high predictive value as renal biomarkers in the pathogenesis of LN, as confirmed by animal validation.

## INTRODUCTION

Lupus nephritis (LN) is an autoimmune-associated kidney disease and one of the primary clinical manifestations of systemic lupus erythematosus (SLE) (*Mok et al., 2023*). LN impacts a significant portion of individuals with SLE. Specifically, it impacted 30∼50% of adults with SLE and up to 60∼70% of pediatric SLE within five years of diagnosis (*Tsai et al., 2023*). The occurrence and progression of LN are pivotal in determining the prognosis of SLE, highlighting the importance in patient management and treatment planning (*Obrisca et al., 2021*). The pathophysiology of LN is still unclear (*Gensous et al.,*

Corresponding authors
Mingyi Zhao,
zhao_mingyi@csu.edu.cn
Qingnan He, heqn2629@csu.edu.cn

*2021*). Various autoantibodies, including ds-DNA, act in the immune system and generate immune complexes that lodge in the kidneys and set off inflammatory responses that continuously harm the kidneys (*Crow, 2023*). Although the use of drugs such as new immune-suppressants and biological agents has improved the prognosis of LN to some extent, 10% to 20% of cases progressed to end-stage renal disease within 5 years after the diagnosis of LN (*Rovin et al., 2021*). Moreover, subclinical renal inflammation can persist even when proteinuria shows improvement (*Omer et al., 2024*). The main reason is the current lack of biomarkers for early diagnosis of LN and corresponding targeted treatment methods (*Yu et al., 2022*). Therefore, it is crucial to thoroughly investigate the pathophysiology of LN, hunt for key biomarkers that accurately reflect the underlying renal damage process and more reliably predict the progression of LN and responses to treatment.

Currently, numerous new biologics targeting the immune system, including B cells, T cells, and cytokines, have emerged and hold potential as therapeutic biomarkers for lupus nephritis (*Fava et al., 2022*). Additionally, in the era of precision medicine, multi-omic techniques offer a novel strategy to identify potential biomarkers for LN (*Chen et al., 2023*), such as microRNAs (*Garmaa et al., 2024*), tRNA-derived small non-coding RNAs (*Yang et al., 2024*), and serum or urinary metabolites (*Guo et al., 2024*). However, there are no identifiable biomarkers for lupus nephritis, and the exact cause of the condition is unknown (*Yu, Nagafuchi & Fujio, 2021*). To identify the kidney biomarkers of lupus nephritis, this research uses bioinformatics to examine the expression microarray data of the glomerulus and tubulointerstitium in LN, then excavate the kidney-specific indicators and possible pathways to discover potential therapeutic targets of LN. Finally, verify the significance of the identified biomarkers by using the MRL/lpr lupus mouse model.

## MATERIALS & METHODS

### Data sources

For the current study, the datasets, namely GSE200306 (*Parikh et al., 2022*) (Platforms: GPL21847 (HG-U133_Plus_2)) have been retrieved from Gene Expression Omnibus (GEO) (http://www.ncbi.nlm.nih.gov/geo/). This dataset has been meticulously curated, encompassing both clinical and histological data for all patients. Initially, the GSE200306 dataset included 58 paired kidney biopsies including both first biopsy and second biopsy from individuals diagnosed with proliferative lupus nephritis. These biopsies were classified as either class III, class IV, or class III/IV+V, in which glomeruli and tubulointerstitium was isolated separately. Our study specifically analyzed only the first biopsy data of LN from this dataset including 34 LN glomerular samples and 45 LN tubulointerstitial samples. Additionally, nine glomerular samples and 10 tubulointerstitium samples from pre-implantation donor kidney biopsies were used as healthy controls (HC). Overall, three clusters from the datasets were delineated for bioinformatic analysis: glomerular cluster (LN_G group, 34 LN and 9 HC), tubulointerstitium cluster pairs (LN_T group, 45 LN and 10 HC), and integrated total renal cluster (LN group, 79 LN and 19 HC).

### Identification of DEGs

The GEO dataset (GSE200306) was downloaded using the "GEOquery" package in R software. Expression data and phenotypic data were extracted from the dataset. First, we utilized the normalizeBetweenArrays function from the "limma" package (*Ritchie et al., 2015*) to normalize the microarray data ensuring that the expression values are comparable across different arrays. Then, a design matrix was created to represent the different groups under comparison. This matrix was essential for fitting the linear model. we analyzed differentially expressed genes (DEGs) in three distinct clusters: total kidney (LN group), glomeruli (LN_G group), and tubulointerstitium (LN_T group) from primary LN patients and healthy controls. DEGs were identified based on the following criteria: adjust. $P < 0.05$ and fold change (FC) $>1$. The "ggplot2" package was employed to visualize the differential DEGs results including volcano plot, heatmap and boxplot. The full code for R can be found in the R code supplemental file.

### Functional and pathway enrichment analysis

We employed the "clusterProfiler" package in R for conducting functional enrichment analysis of the differentially expressed genes (DEGs) based on Gene Ontology (GO, http://www.geneontology.org) and the Kyoto Encyclopedia of Genes and Genomes (KEGG, https://www.kegg.jp). In the context of the GO analysis, we identified and explored three distinct categories: biological process (BP), cellular component (CC), and molecular function (MF) (*The Gene Ontology C, 2019*), shedding light on the biological processes associated with these DEGs. We also explored potential signaling pathways using the KEGG analysis.

### Vene plots and PPI networks construction

To investigate the core differentially expressed genes (DEGs) in primary LN across different renal compartments, we analyzed the total renal group (LN group), the glomeruli parts (LN_G group), and the renal tubulointerstitium (LN_T group). Using an online tool (https://www.xiantaozi.com/), we generated Venn plots to visualize the overlaps and distinctions among these groups. To evaluate the relationships between the DEGs in the dataset, we established a protein-protein interaction (PPI) network using the Search Tool for the Retrieval of Interacting Genes (STRING v11.5) (*Szklarczyk et al., 2021*). Subsequently, we integrated the results obtained from the STRING database into Cytoscape software (v3.9.1) to visualize the protein-protein interactions (PPIs) within the statistically significant DEGs.

### NephroSeq datasets verification

NephroSeq database (https://nephroseq.org/) is an free online platform designed for the exploration and analysis of gene expression data specifically related to nephrology. NephroSeq contains a vast collection of gene expression datasets from human nephropathy studies, such as diabetic nephropathy, IgA nephropathy, lupus nephritis. NephroSeq was used in our study to verify the changing trends and differences in the expression of seven DEGs identified from the GSE200306 dataset. By leveraging the extensive datasets available in NephroSeq, we were able to cross-validate our findings and gain additional insights into

the behavior of these genes in LN datasets. A heat map displayed the $p$-values for the up and down expression of the seven DEGs, providing a clear visual representation of the statistical significance of gene expression changes.

## ROC curve for screening the key biomarker

Using the outcome indicators in original sample data of GSE200306 to analyze the most important DEGs of glomeruli parts (LN_G group), and renal tubulointerstitium parts (LN_T group) respectively in primary LN by prism (version 8.0.0). And we also plotted ROC curves of common DEGs in the LN_G group and LN_T group to identify more diagnostically valuable common DEGs. When the value of Area Under Curve (AUC)>0.8, it means that the performance of the classifier is better, and the prediction result of the classifier is more reliable.

## Animals experiment in MRL/lpr mice

Purchased 8-week-old SPF-level female MRL/lpr lupus mice ($n = 3$) and 8-week-old SPF-level female healthy control mice C57BL/6 ($n = 3$) from the SPF Biotechnology Co., Ltd. Mice were fed standard laboratory chow and supplied drinking water ad libitum in Department of Zoology, Central South University. All animal experiments were performed in accordance with the Guidelines for the Care and Use of Laboratory Animals published by the US National Institutes of Health (NIH). No animal intervention experimental study was carried out in this study. The criteria for euthanizing animals in our studies include: (1) Severe illness or injury that cannot be effectively treated; (2) significant and unrelievable pain or distress; (3) failure to thrive or meet basic welfare requirements; (4) development of unexpected complications that compromise the animal's well-being; (5) completion of the experimental procedures or endpoints. If any of the aforementioned situations arise, we will euthanize the animals using carbon dioxide ($CO_2$) euthanasia. Fill $CO_2$ at an even rate to achieve a $CO_2$ replacement rate of 30% to 70% of the container volume per minute, then animals lose consciousness and die within 2 to 3 min. Executed the mice when 25 weeks old and collected plasma, urine, and kidney samples for further analysis. Plasma samples were used to measure ds-DNA concentration, urine samples were analyzed for protein quantification, and kidney samples were used to assess mRNA and protein expression levels. Sampling procedures were conducted under light anesthesia using isoflurane to minimize discomfort and ensure the welfare of the animals. No animals survived at the conclusion of the experiment. The animal experiments mentioned above have been approved by Central South University's Animal Welfare Ethics Review (Number: CSU-2022-0133).

## Validation of key DEGs in MRL/lpr mice

The Bradford method was used to detect routine urine protein quantification. Plasma ds-DNA concentration was detected by mouse anti-double stranded DNA antibody (IgG) ELISA Kit (CSB-E11194m; CUSABIO). RT-qPCR was used to detect the mRNA expression in kidney tissue of five key DEGs screened in GSE200306. Primers of RT-qPCR were designed using primer 3 (https://primer3.ut.ee/), and the primer sequences are shown in Table S1. The reverse transcription was performed using HiScript II Q RT SuperMix for

**Table 1** Expression of seven common DEGs in the total renal LN group of GSE200306.

| Probe_id | log2FC | Mean Expression | P value | adj. P value | DEG |
|---|---|---|---|---|---|
| EGR1 | −2.006777115 | 7.15881716 | 9.73E−12 | 4.84E−09 | Down |
| MME | −1.847965404 | 7.513895773 | 2.14E−11 | 5.32E−09 | Down |
| PTPRC_all | 1.461323403 | 3.507084548 | 4.17E−10 | 6.91E−08 | Up |
| RORC | −1.468197324 | 5.374135777 | 1.91E−09 | 1.90E−07 | Down |
| MX1 | 1.471260078 | 6.451864327 | 6.18E−07 | 3.41E−05 | Up |
| ZBTB16 | −2.531346993 | 5.942123074 | 4.07E−06 | 0.00014452 | Down |
| FKBP5 | −1.167577295 | 5.250672116 | 0.000356437 | 0.003615288 | Down |

qPCR (Vazyme, R223), and the RT-qPCR process was carried out with ChamQ Universal SYBR qPCR Master Mix (Q711; Vazyme). The protein expression difference of DEGs in kidney was observed by immunohistochemistry, in which 10 different parts of each stained section were selected under a 40x optical microscope, and the average optical density was calculated using ImageJ2 software (version 2.3.0). Experimental antibodies were purchased from Servicebio. Statistical analysis was conducted using GraphPad Prism software (version 8.0.0). Unpaired $t$-test analysis was utilized for between-group comparisons, with statistical significance set at $P < 0.05$.

# RESULTS

## Identification of DEGs From the dataset

Our study utilized the gene expression profiles from the GSE200306 dataset obtained from the GEO database, which were originally submitted by *Parikh et al. (2022)*, with platforms GPL21847 (HG-U133_Plus_2). After obtaining the original data, R software was used to control the quality of the data, which showed that the data of the LN group and the control group had good uniformity (Fig. 1A). A total of 497 genes were detected between the two groups, and the volcano plots showed significant differences in the minority (Fig. 1B). In the total renal of primary LN groups (LN group), we identified 17 hot DEGs (Fig. 1C). In comparison, 25 DEGs were discovered in the glomeruli group (LN_G group), and 24 DEGs were identified in the renal tubulointerstitium group (LN_T group) when comparing with control groups (Figs. 1D and 1E). These findings underscore the distinct molecular signatures present in different renal compartments.

## Key DEGs and function analysis

Using Venn plot analysis, a total of seven significantly differential genes were shared among the three comparison groups (Fig. 2A). In these three groups of comparisons, the changes in the above seven DEGs are consistent, in which *PTPRC and MX1* were the up-regulated genes, while *EGR1, MME, RORC, ZBTB16, FKBP5* were the down-regulated DEGs. Table 1 showed the detail changes of the seven common DEGs in the total renal group (LN group). We used the STRING tool to discover the protein interactions among the common differential DEGs, as a result, *PTPRC, MME* and *MX1* occupied the most central position in the network (Fig. 2B).
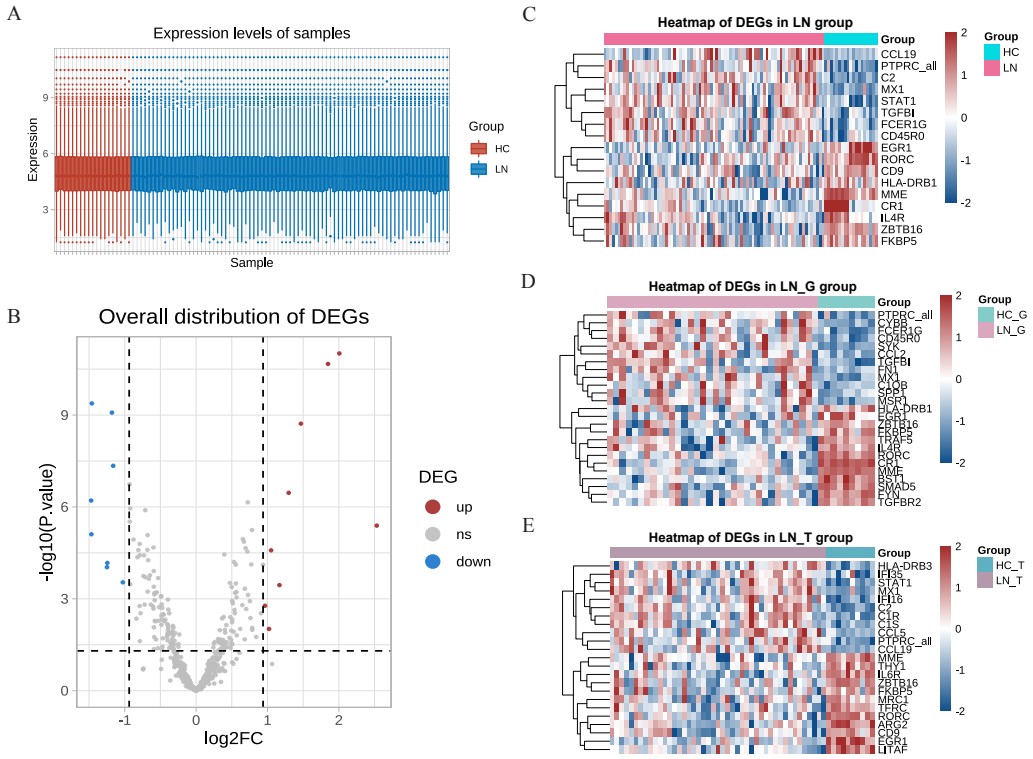

**Figure 1  DEGs identified from the GSE200306 dataset.** (A) Quality control of the original data in the total renal of LN group and HC group. The red shows samples of HC group, the blue shows samples of LN group. *Y*-axis shows the minimum, first quartile, median, third quartile, and maximum values of detected genes expressions. (B) Volcano plots showing the significant differential genes in the total renal of LN group and HC group. (C–E) Heatmap of significant DEGs in three clusters, including total renal (C), glomeruli (D) and tubulointerstitium (E) between primary LN patients and healthy controls.

We performed functional enrichment analysis of DEGs between the LN and HC groups concentrating on KEGG and GO pathways. KEGG showed hematopoietic cell lineage, TH1/2/17 cell differentiation was the most related pathway in the LN pathogenesis (Fig. 2C). GO analysis demonstrated that T cell differentiation, lymphocyte differentiation and mononuclear cell differentiation participated in the key parts of the biological process of LN (Fig. 2D).

## NephroSeq datasets analysis

Three datasets in the NephroSeq database (https://nephroseq.org/) from two published articles (*Berthier et al., 2012*; *Peterson et al., 2004*) and not yet published by ERCB in 2018/04/01 (see Data Availability section) were included in the analysis. The three databases contain both glomerular and tubulointerstitium gene clusters of lupus nephritis, in which total 62 subgroups from six Lupus datasets were analysed in our study, including Berthier Lupus Glom ($n = 46$), Berthier Lupus Mouse Kidney ($n = 68$), Berthier Lupus TubIntn ($n = 47$), ERCB Lupus TubInt ($n = 41$), ERCB Lupus Glom ($n = 32$) and Peterson Lupus Glom ($n = 31$) (Tables S2 and S3). In the comparison of 62 subgroups, the up-regulated gene

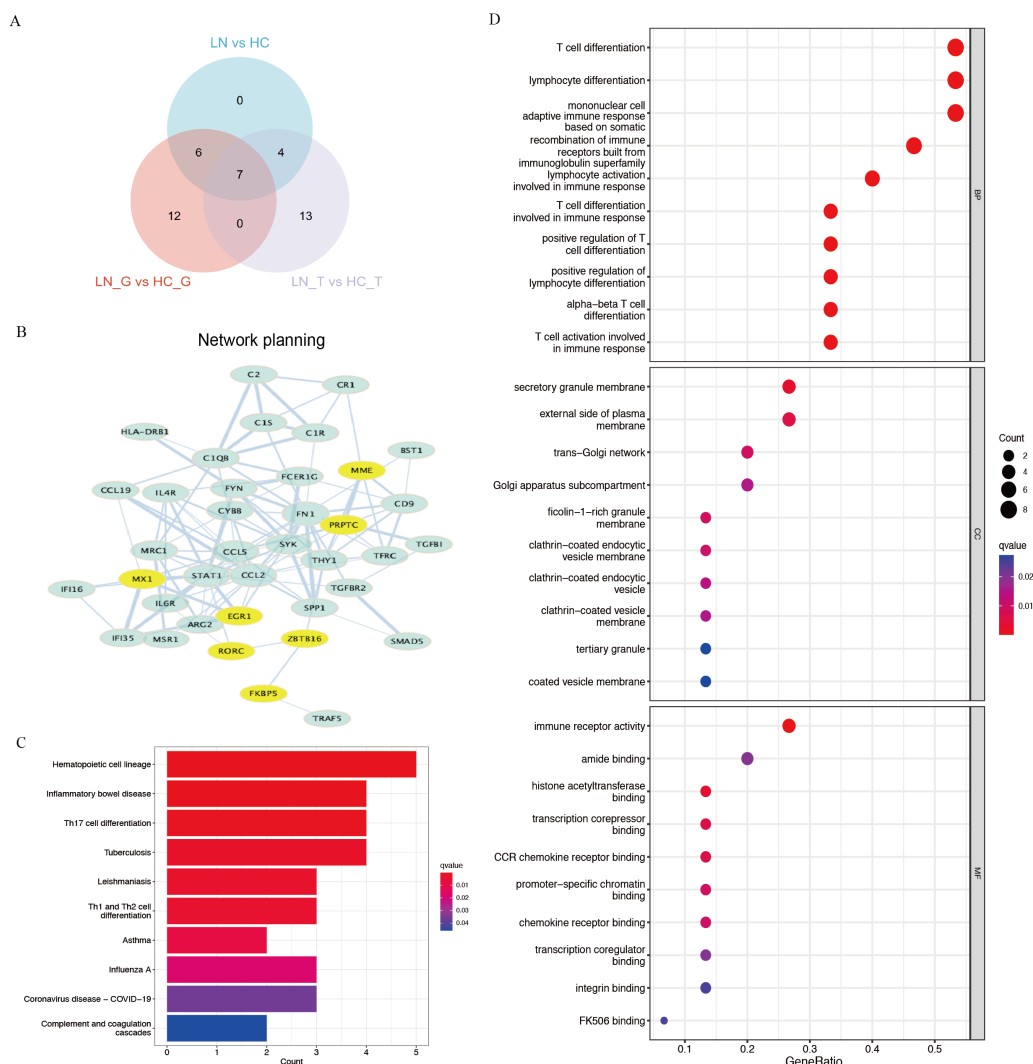

**Figure 2 Key DEGs and function analysis by KEGG and GO pathway.** (A) Venn diagram of the three clusters (LN_G group, LN_T group, and LN group). (B) PPI networks of total DEGs in the three clusters, the yellow shows the common DEGs in all three clusters. (C) KEGG pathway analysis of DEGs in the total renal LN group. (D) GO pathway analysis of DEGs in the total renal LN group.

(*PTPRC, MX1)* and the down-regulated DEGs (*EGR1, MME, RORC, ZBTB16, FKBP5*) were verified to have mostly consistent change trends in all Lupus Nephritis *vs.* Normal Kidneys group with the results of GSE200306 (Fig. 3). In addition, proteinuria status and LN pathological class also showed significantly different gene expression, especially in *PTPRC* and *MME*.

## ROC curve for screening the key biomarker

The diagnostic ROC curves of DEGs were analyzed based on glomerulus and tubules respectively (Table S4). Among these, the seven common DEGs in the glomeruli group (LN_G) and the renal tubulointerstitium group (LN_T), which were identified in the

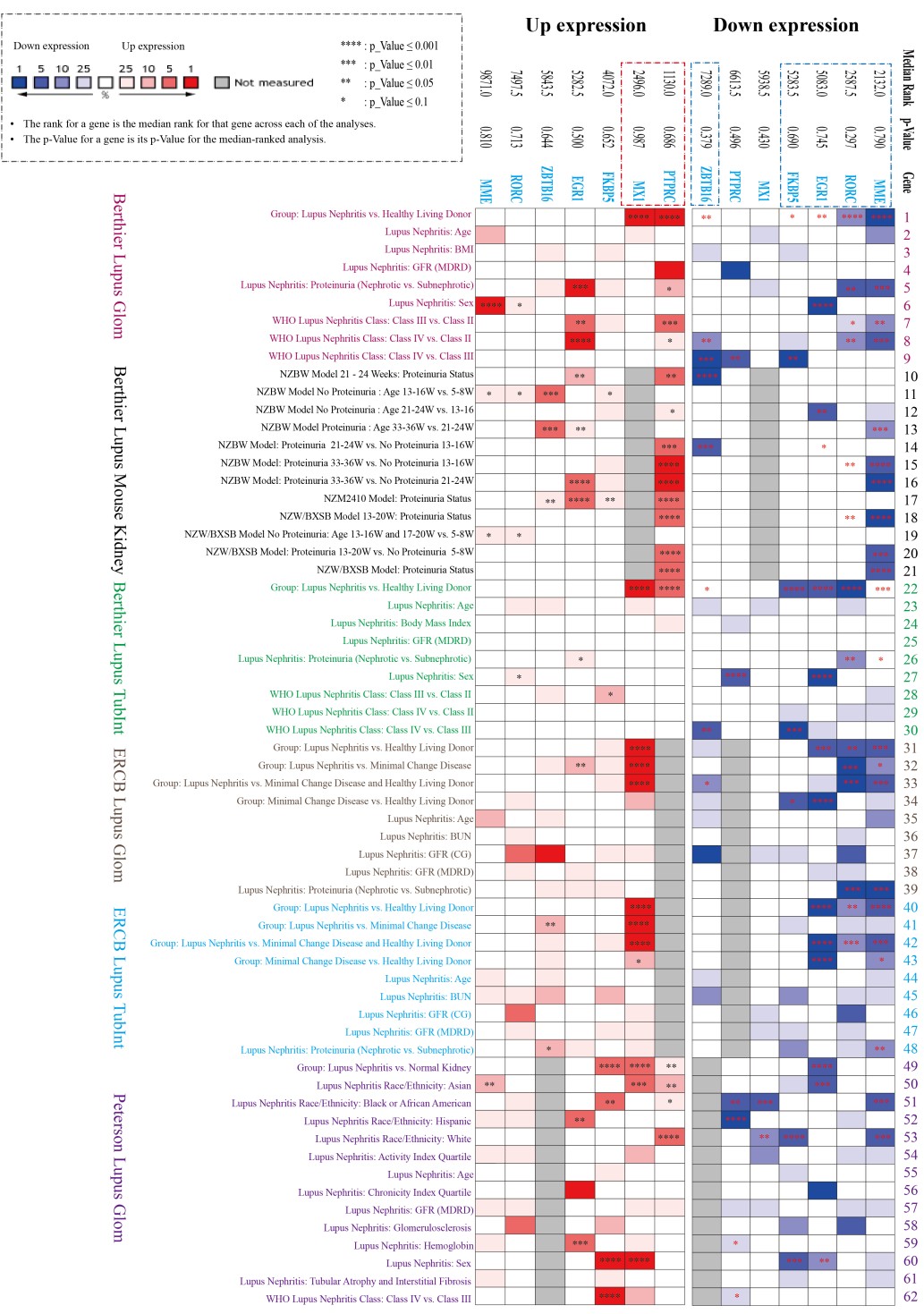

**Figure 3  Seven key DEGs verified in the NephroSeq database.** Three NephroSeq datasets (https://nephroseq.org) with both glomerular and tubulointerstitium gene clusters of lupus nephritis were included in the analysis. Red indicates up-regulated expression, and blue indicates the down-regulated expression (Tables S2 and S3).

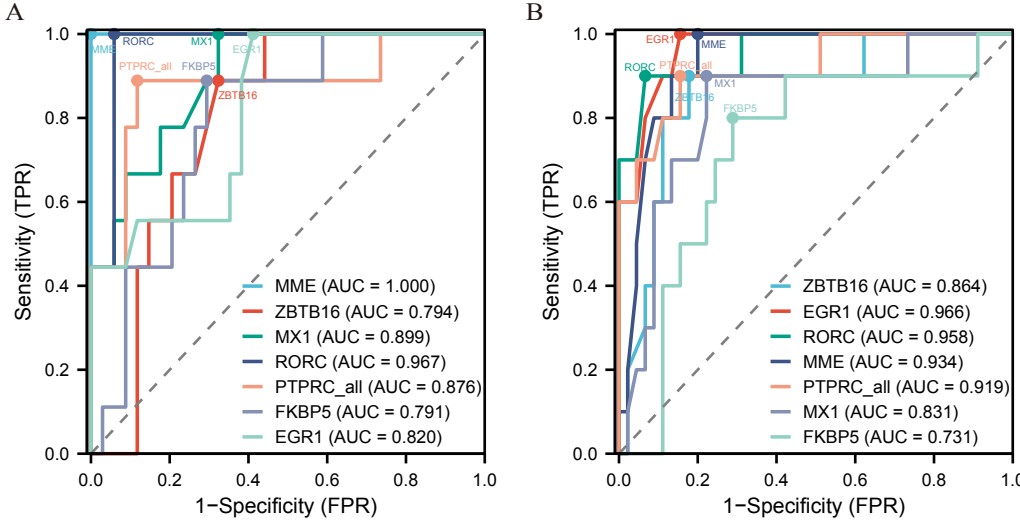

**Figure 4  The key renal biomarkers screened by ROC curve analysis.** The analysis data comes from the outcome indicators in original sample data of GSE200306. AUC >0.80 shows a higher predictive value.

Venn diagrams and verified in the NephroSeq database, were used to plot ROC curves and identify the diagnostic renal biomarkers for LN (Fig. 4). Consequently, five common DEGs including *PTPRC, MX1, EGR1, MME* and *RORC*, were found to be more promising biomarkers in both glomerulus (Fig. 4A) and tubulointerstitium (Fig. 4B) with AUC values greater than 0.80.

## Key renal biomarkers verified by MRL/lpr mice

Using SPF-level female MRL/lpr lupus mice ($n = 3$) and female C57BL/6 ($n = 3$) to make a verification for the five key DEGs screened by ROC analysis. The MRL/lpr mice exhibited notable lymph node enlargement and characteristic rashes starting at 15 weeks of age (Fig. 5A). Additionally, an increased urinary albumin/creatinine ratio (UACR) was observed using the Bradford method. At 25 weeks, both groups were sacrificed, and plasma and kidney samples were collected for further analysis.. The results revealed that MRL/lpr mice showed significantly higher plasma ds-DNA concentrations and UACR compared to the C57BL/6 mice (Figs. 5B and 5C).

Through RT-qPCR detection, the five DEGs showed consistent changes with the analysis results of the GEO dataset. The mRNA expressions of *PTPRC* and *MME* were significantly different between the MRL/lpr group and the C57BL/6 group, and the expression of *PTPRC* was up-regulated while the expression of *MME* was down-regulated (Fig. 6A). Immunohistochemical staining analysis of renal histopathological sections also showed changes consistent with mRNA levels, and *PTPRC, MME* and *MX1* also had significant differences at the protein level (Fig. 6B).

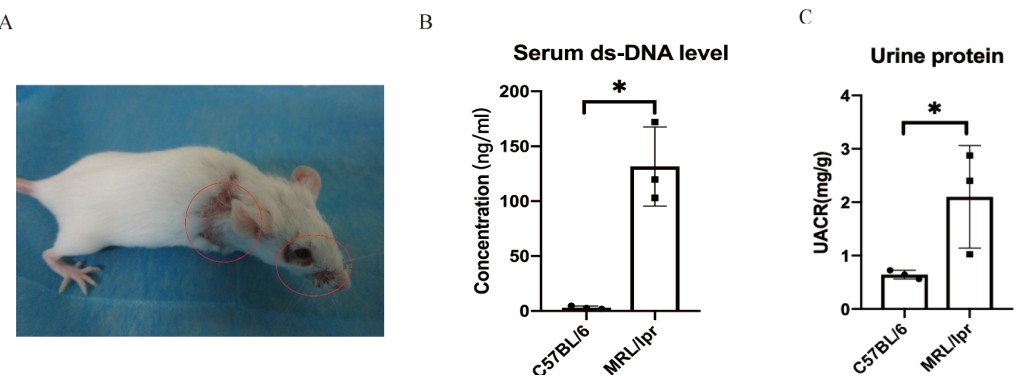

**Figure 5 Representative clinical features of MRL/lpr mice.** (A) The characteristic rashes of MRL/lpr mice (circled in red). (B) In 25-weeks old MRL/lpr mice, the serum ds-DNA levels were significantly higher than C57BL/6 mice. (C) Increased urinary albumin/ creatinine ratio (UACR) in MRL/lpr mice ($p < 0.05$).

# DISCUSSION

Lupus nephritis has poor prognosis, the mechanism of which is still unclear, and there are no characteristic biomarkers. In this study, bioinformatics analysis of GEO database was used for finding the renal biomarkers in lupus nephritis. Using the differential gene analysis of different cluster combinations of total kidney, glomerulus, and tubulointerstitium, seven core genes were focused and verified by the lupus mice model. The seven hot differential genes were verified to have consistent change trends using three datasets from NephroSeq database. Through the ROC analysis, five DEGs (*PTPRC, MX1, EGR1, MME* and *RORC)*, were found to be good biomarkers in both glomerulus and tubulointerstitium. Then, the mRNA and protein expression of the five genes verified in the kidney of MRL/lpr mice, showed that *PTPRC* and *MME* were found to be significantly different in the renal of lupus mice.

*PTPRC*, formally known as protein tyrosine phosphatase receptor type C, is a transmembrane glycoprotein and a member of the protein tyrosine phosphatase family. It is often referred to as CD45 and is recognized as a leucocyte common antigen. *PTPRC* is widely expressed on nearly all hematopoietic cells, except for mature erythrocytes, and plays a crucial role as a regulator in the activation of T and B cell antigen receptors (*Al Barashdi et al., 2021*). Phosphatase activity and methylation levels of *PTPRC* found to be different in SLE patients (*Imgenberg-Kreuz et al., 2018*; *Szodoray et al., 2016*). In a co-aggregation gene study involving autoimmune diseases such as primary Sjogren's syndrome, systemic lupus erythematosus, and rheumatoid arthritis, PTPRC was found to exhibit significant differential protein expression in all three disease types. In our finding, the KEGG and GO pathway suggests *PTPRC* may have a close relationship between autoimmune diseases and the regulation of T cell activation and the T cell receptor signaling pathway (*Wang et al., 2020*). *PTPRC* in our study was also shown to be a key biomarker with an upward trend in the kidney of LN, predicted located in membrane and in tracellular, and highly expressed in glomerular mesangial area and parietal epithelial cells. The results of KEGG and GO

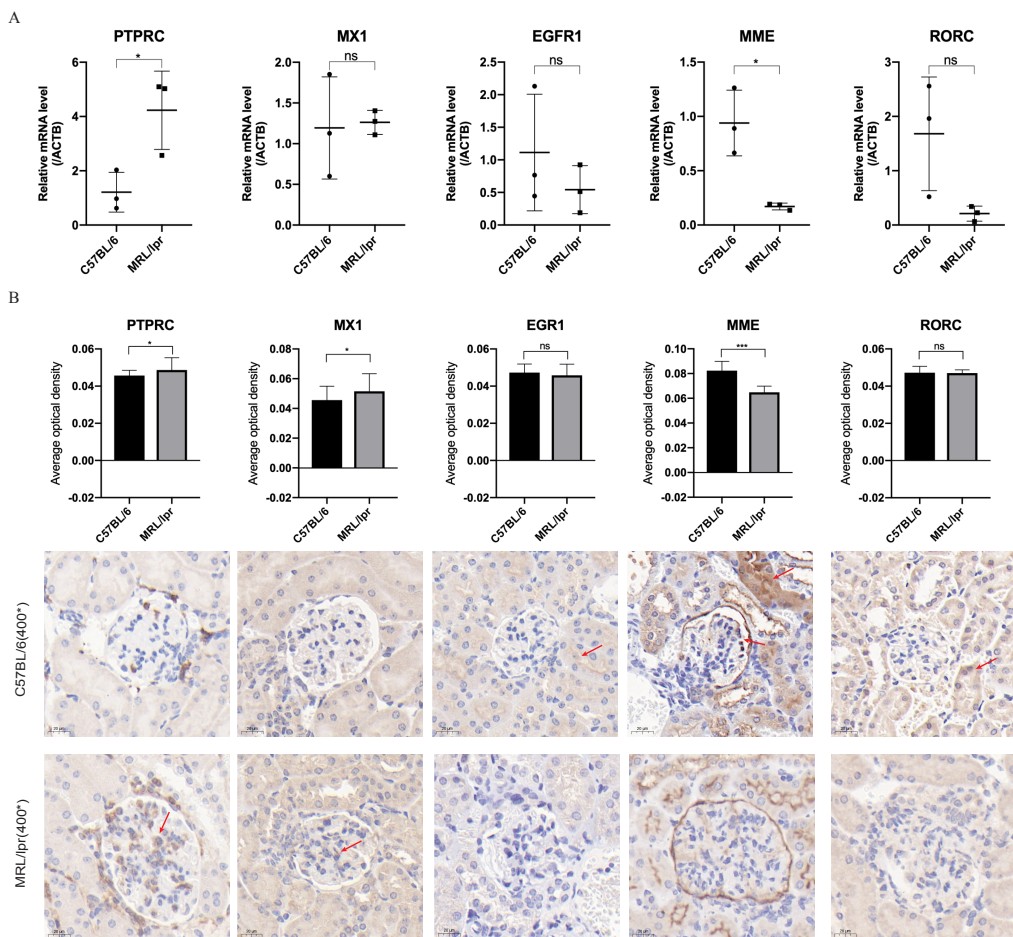

**Figure 6 Five key biomarkers verified in the kidney of MRL/lpr and C57BL/6 mice.** (A) mRNA expression of five key renal biomarkers screened by ROC curve (three technical replicates for three biological replicate seach group). (B) Immuno-histochemical staining analysis of the five key renal biomarkers in renal histopathology of MRL/lpr and C57BL/6 mice (three biological replicate seach group, 10 different parts of each stained section under 400* microscope were selected for the average optical density calculation; positive areas are indicated with red arrows).

pathway analysis showed hematopoietic cell lineages, especially T cell differentiation, are critically important in the pathogenesis in the lupus nephritis, thus suggesting *PRPTC* and its modification may be a potent target for the immune cell regulation in LN, while the specific regulatory mechanism needs to be further studied.

MME, also named membrane metalloendopeptidase, neutral endopeptidase (NEP), neprilysin, and so on, is a type II transmembrane glycoprotein (*Cheng et al., 2020*). MME is present on glomerular epithelium and the brush border of the proximal tubules (*Zhang et al., 2011*). The expression was higher in normal kidney tissue, and a downward trend was found in MRL/lpr mice. The down-regulation of MME in esophageal squamous cell carcinoma tumor tissues is correlated to poorer prognosis of the patients *via* FAK-RhoA axis (*Li et al., 2019*). Overexpression of MME inhibits

substance P stimulation of cholangiocarcinoma growth (*Meng et al., 2014*). Our study found that MME screened by ROC curve analysis showed good predictive ability in both glomeruli and tubulointerstitium. Further research on the cell types and related pathways of MME expression differences in LN may have a certain possibility of explaining the disease occurrence and prognosis prediction of LN.

Even though MX1, EGR1 and RORC were tested to be DEGs in the glomeruli and tubulointerstitium of primary lupus nephritis patients, in MRL/lpr mice, the difference was not being detected differentially. The aforementioned reasons could stem from inconsistencies in sample batches, a limited number of animal verification samples, and variations in transcriptome and protein levels (*Chung et al., 2021*). MX1 (MX Dynamin Like GTPase 1), can regulate cytokine signaling in the immune system by GTP binding and GTPase activity (*Haller et al., 2015*). Differences were also found in transcriptome data analysis of other LNs study (*Wang et al., 2022*; *Li et al., 2022*). In our immune-histochemical detection results of lupus mice, MX1 also showed a tendency to increase, Therefore, the importance of mx1 in the pathogenic mechanism of lupus nephritis cannot be ignored. Early growth response 1 (EGR1) is an "immediate early" transcription factor capable of regulating the expression of numerous downstream genes involved in cell differentiation, proliferation, and the inflammatory response (*Chen et al., 2022*). In cases of acute kidney injury, the rapid and transient induction of EGR1 serves a renoprotective role in facilitating kidney repair (*Chen et al., 2022*). RORC (RAR Related Orphan Receptor C), also known as Nuclear Receptor ROR-Gamma, is a transcription factor for Th17 cells, participating in the Th17 cell differentiation (*Hall et al., 2022*; *Croft et al., 2022*). Lupus nephritis is a complex immune system disease where many important genes and their pathways play key roles in its pathogenesis. Therefore, the genes identified in this study require further in-depth research.

## CONCLUSIONS

This study identified seven potential differential biomarkers through bioinformatics analysis of the GEO database and demonstrated, through animal validation, that *PTPRC* and *MME* may be core kidney biomarkers involved in the pathogenesis of LN. In this study, we examined differential gene expression in the glomerulus and tubulointerstitium separately. We utilized R to integrate the expression data from both compartments, followed by further analysis of DEGs and pathway enrichment. Although limited to one GEO dataset, we also searched and analyzed related transcriptome data in NephroSeq database of LN as a reference. In addition, due to the difficulty of obtaining clinical samples, samples from patients with LN were not used for verification, which is a great limitation of this study. However, it is hoped that through the verification results of LN model mice, it can further explore the pathogenesis of LN and have certain predictive value as a renal marker for early detection and treatment of LN.

### Funding

This study was funded by Fundamental Research Funds for the Central Universities of Central South University (CX20230378). The funders had no role in study design, data collection and analysis, decision to publish, or preparation of the manuscript.

### Grant Disclosures

The following grant information was disclosed by the authors:
Fundamental Research Funds for the Central Universities of Central South University: CX20230378.

### Competing Interests

The authors declare there are no competing interests.

### Author Contributions

- Min Wen performed the experiments, analyzed the data, prepared figures and/or tables, authored or reviewed drafts of the article, and approved the final draft.
- Marady Hun analyzed the data, prepared figures and/or tables, authored or reviewed drafts of the article, and approved the final draft.
- Mingyi Zhao conceived and designed the experiments, authored or reviewed drafts of the article, and approved the final draft.
- Qingnan He conceived and designed the experiments, authored or reviewed drafts of the article, and approved the final draft.

### Animal Ethics

The following information was supplied relating to ethical approvals (*i.e.*, approving body and any reference numbers):

The animal experiments mentioned above have been approved by Central South University's Animal Welfare Ethics Review (Number: CSU-2022-0133).

### Data Availability

The raw measurements are available in the Supplemental File. The data is available at Genbank (GSE200306) and NephroSeq (Tables S2 and S3).

ERCB dataset summaries can be found at: ERCB Lupus Glom Dataset Summary. Available at https://nephroseq.org/resource/ui/component/dataset.html[balance] (requires registration). ERCB Lupus TubInt Dataset Summary. Available at https://nephroseq.org/resource/ui/component/dataset.html[balance] (requires registration).

### Supplemental Information

Supplemental information for this article can be found online at http://dx.doi.org/10.7717/peerj.18070#supplemental-information.

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
