# Peer review of "MME and PTPRC: key renal biomarkers in lupus nephritis"

_PeerJ, doi:10.7717/peerj.18070_

## Round 0.1 · original submission · Major Revisions

The manuscript requires major revisions. The reviewers' comments provide valuable suggestions that need to be addressed. Please ensure you respond to the feedback from all three reviewers. Reviewer 3 has provided comments in an annotated manuscript. Please download this file and address each comment accordingly.

·

Basic reporting

This study bioinformatically identified seven renal biomarkers in Lupus Nephritis using a GEO dataset. Five biomarkers were bioinformatically validated using three datasets from NephroSeq, another public dataset. Two biomarkers were experimentally validated via messenger RNA and protein levels in the kidneys of MRL/lpr mice. Publication’s introductory section provides essential context, ensuring readers grasp the topic from outset. Following are the issues found with basic reporting:
1) To ensure clear understanding by an international audience, spelling mistakes should be fixed and enhancements should be made to the English language within the manuscript's text. Specific areas for improvement, such as lines 41-42, 90-93, 152-154, 186-190, 238, 256-259 where the current wording poses comprehension challenges.
2) Acronyms for the three groups are introduced for the first time in lines 91-92. I suggest mentioning them in the “Data sources” section (lines 70-72).
3) In Table 1, the legend doesn't specify which of the three clusters are represented in columns 2 to 6.
4) In Figure 1A, the legend does not clarify which genes are depicted on the Y-axis. Additionally, the X-axis lacks labels for sample identifiers.
5) In Figure 1B, the legend doesn't indicate that the volcano plot illustrates differentially expressed genes between the total renal group and the control.
6) In Figures 1C, 1D, and 1E, the color legend does not clearly indicate the metric represented by the heatmap. Additionally, the use of one common color between the group color schema and the heatmap metric color schema is causing confusion in interpretation.
7) In Figure 2A, the choice of color scheme, particularly for the LN_T vs HC_T section, makes it difficult to discern. While the figure illustrates the number of common differentially expressed genes between different clusters, it does not indicate whether the log fold change and direction of change also align.
8) Readability of Figure 2B is compromised due to the small font size.
9) In Figure 2D, some of the Y-axis labels are notably challenging to read.
10) In Figure 4, the ROC plots should have a title. The legend does not specify which dataset the ROC curves are based on. Ideally, these ROC plots should be derived from an independent validation dataset since the GEO dataset was already used to identify the seven biomarkers.

Experimental design

Overall this study presents a nicely crafted research question, but here are some of the issues with methodologies in the study:
1) Lines 68 - 72: It is unclear if the three clusters were balanced for confounding factors such as gender and age.
2) Lines 68 - 72: It is unclear why the LN_G cluster comprises only 34 LN samples when there were a total of 70 glomerular samples from LN patients.
3) Lines 68 - 72: It is unclear why the LN_T cluster comprises only 45 LN samples when there were a total of 92 tubulointerstitial samples from LN patients.
4) Lines 68 - 72: It is unclear how the total renal cluster encompasses 19 healthy controls, considering there were only a total of 10 pre-implantation donor biopsies.
5) Lines 68 - 72: It is uncertain whether each cluster contains LN samples from distinct patients, or if patient samples were repeated within a cluster or across clusters.
6) Lines 74 - 80: I suggest that this section should have a description on methodologies for quality control checks.
7) Lines 74 - 80: The authors state the utilization of the limma package but the specific functions employed from the package have not been disclosed. Without these details it remains unclear whether the choice of this package and functions effectively address all variances.
8) Line 80: The specified R code is missing from supplementary files. Sharing the code would enhance transparency and facilitate the reproducibility of your work.
9) Lines 99 - 103: I suggest that this section should include a short description of the gene expression dataset, including whether it was obtained through microarray or NGS techniques.
10) Lines 99 - 103: I suggest that this section should have a short description on sample set sizes for each cluster.
11) Lines 99 - 103: It is unclear if any normalization was done with this dataset.
12) Lines 99 - 103: It is unclear if any quality control filtering or contamination checks were done with this dataset.
13) Lines 99 - 103: I suggest that this section include a concise summary outlining the methodologies and objectives behind the 62 different analyses referenced in Figure 3 and Supplementary Table 3.
14) Lines 105 -109: It is unclear which dataset was used for this analysis. Ideally, it should be based on an independent dataset separate from the GEO dataset, as the latter was already utilized for selecting biomarkers.
15) Lines 105 - 109: Important parameters like cross validation folds remain unclear and are not described in this section.
16) Lines 108 - 109: The authors mention a threshold of 0.8, but it remains unclear why this specific threshold was selected.
17) Lines 131 - 143: It is unclear if any normalization was done with the dataset mentioned in this section.

Validity of the findings

Even the authors acknowledge that biomarkers were not validated with clinical samples. Additionally, mice experimental dataset size is very small. These limitations underscores the need for cautious interpretation of the findings, as the sample size may affect the generalizability and robustness of the results. Following are the issues with the results section:
1) Lines 149 - 150: These findings indicate that there is no batch effect but it is unclear from the figure and the text if all or certain genes were used for this analysis.
2) Lines 170 - 176: I suggest that this section should have more description on the findings from the 62 different analyses mentioned in figure 3 and supplementary table 3.
3) Lines 178 - 182: It is unclear if the authors checked AUC for LN vs control with confounding factors like age, gender etc from the study’s metadata.
4) Lines 178 - 182: It is unclear if any checks were done. Typically group assignments are randomly shuffled and the AUC in such cases should be around 0.5.
5) Lines 178 - 182: These findings are valid only if the analysis was done with independent validation dataset, i.e, NephroSeq and not with GEO dataset.
6) Lines 184 - 197: It is unclear which experiments were conducted specifically with plasma samples and which were conducted with kidney samples.

Additional comments

Despite the study design constraints, the study offers valuable insights within the scope of its available data, highlighting areas for potential future research with larger datasets to further validate and extend its findings.

Reviewer 2 ·

Basic reporting

The present manuscript entitled “MME and PTPRC: key renal biomarkers in lupus 2 nephritis” that outlines an important contribution to the literature in the field of Auto immune disease. Overall, the paper is technically sound, thoughtful and generally supports their conclusions. But there is no sufficient information about the novelty and clinical relevance of the current research.

Experimental design

Experimental design of the current manuscript is suitable for this journal.

Validity of the findings

Results: The table quality is looking fine for the publication.

Additional comments

• Discussion: Authors should be discussed more with new insight which will provide more strength to the current research in the field of arthritis. A more mechanistic presentation and explanation would better communicate the take-home message.
• Over all after addressing the comments the current manuscript has significance for the publication in this esteemed journal.

Cite this review as

Reviewer 3 ·

Basic reporting

• Clarity and Structure: The article is clearly structured, with a defined background, methods, results, and conclusions sections. The language is generally clear and professional.
• Literature References: While the article provides a background on LN, it lacks citations to previous work that could provide context and support for the study’s objectives. It would benefit from a more thorough review of existing literature on LN biomarkers.

Experimental design

• Research Question: The research question is well-defined, aiming to identify renal biomarkers for LN through bioinformatics analysis.

Validity of the findings

Validation using MRL/lpr mice adds credibility to the findings. It confirms the relevance of the identified biomarkers in a biological context, which is crucial for translational research.

Annotated reviews are not available for download in order to protect the identity of reviewers who chose to remain anonymous.
Cite this review as

---

## Round 0.2 · Minor Revisions

Please address the final minor issues raised by the reviewer 1.

·

Basic reporting

The revised manuscript now exemplifies good basic reporting following comprehensive integration of peer review feedback. Noteworthy improvements include the adoption of clear, unambiguous, and professional English language throughout the manuscript, ensuring enhanced readability and comprehension. Additionally, the figures have been refined to uphold relevance, quality, and clarity, with meticulous labeling and comprehensive descriptions that effectively support the findings presented. These enhancements collectively underscore the manuscript's commitment to meeting standards of basic reporting in PeerJ.

Experimental design

There are still some pending concerns regarding the methodology employed for ROC plots. Specifically, it seems likely from the responses that the differential expression analysis was performed prior to making the ROC curves. Ideally, this analysis should be conducted within the cross-validation folds to prevent information leakage. Information leakage occurs when information from outside the training dataset is used in model building, leading to overly optimistic performance estimates. Performing differential expression analysis within each cross-validation fold ensures that the model is evaluated on truly independent data, providing a more accurate and unbiased assessment of its performance.

Validity of the findings

The findings in the manuscript are promising, with results from independent datasets like NephroSeq and a mice model reinforcing the biomarkers identified from the GEO dataset. These additional validations lend credibility to the biomarkers' potential significance. However, concerns remain regarding the ROC curve results. Since it is likely that the differential expression analysis was not performed within the cross-validation folds. This methodological flaw can lead to inflated performance metrics, casting doubt on the robustness of the ROC curve results.

Additional comments

The revised manuscript demonstrates significant improvements in basic reporting. This manuscript can be published based on its findings from NephroSeq and the mice model, as they reinforce the biomarkers identified from the GEO dataset. However, the ROC curve analysis should not be relied upon due to methodological concerns.

Reviewer 2 ·

Basic reporting

The present manuscript can be accepted now in the present form.

Experimental design

Experimental plan is fine

Validity of the findings

Findings of the present study is looking fine.

Additional comments

Thank for inviting me for review the current manuscript.

Cite this review as

Reviewer 3 ·

Basic reporting

The revisions have significantly improved the clarity, rigor, and overall quality of the manuscript. The authors have provided clear and detailed responses to each comment, and the changes made are appropriate and enhance the manuscript's contribution to the field.

Experimental design

The authors have provided additional details and justifications regarding their methodology, which now offers a clear and reproducible approach.

Validity of the findings

The presentation of data has been enhanced, including more comprehensive tables and figures, which facilitate a better understanding of the results

Cite this review as

---

## Round 0.3 · Minor Revisions

Authors have mentioned in their rebuttal letter that "We acknowledge the limitations of the current ROC curve analysis and will revise the methodology accordingly"; however, I did not see any tracked changes in the revised manuscript for the same. Please include changes in the tracked changes manuscript as you have mentioned in the rebuttal letter.

---

## Round 0.4 · accepted · Accept

Authors have addressed all of the reviewers' comments and this manuscript is ready for publication.